# The Expression of *ELF4*-Like Genes Is Influenced by Light Quality in Petunia

**Naoya Fukuda [1,\*], Tomohiro Suenaga [1], Erika Miura [2], Atsuko Tsukamoto [1] and Jorunn E. Olsen [3]**

[1]    Graduate School of Life and Environmental Sciences, University of Tsukuba 1-1-1 Tennodai, Tsukuba, Ibaraki 305-8572, Japan; suenaga.tomohiro@seiwa-ltd.co.jp (T.S.); ako28@shaw.ca (A.T.)

[2]    College of Agro-Biological Sciences, Tsukuba 1-1-1 Tennodai, Tsukuba, Ibaraki 305-8572, Japan; erika.iwt5@gmail.com

[3]    Department of Plant Sciences, Faculty of Biosciences, Norwegian University of Life Sciences, N-1432 Ås, Norway; jorunn.olsen@nmbu.no

[\*]    Correspondence: fukuda.naoya.ka@u.tsukuba.ac.jp; Tel.: +81-29-853-2547 or +81-29-853-6205

**Abstract:** The signals from photoreceptors modify plant morphogenesis and regulate the timing of flowering. In the long-day plant petunia, flowering is accelerated under blue (B) and white (W) light compared to red (R) light. In *Arabidopsis thaliana* L., *ELF* genes are involved in circadian clock-associated regulation of flowering under different light conditions. In this study, we aimed to assess the involvement of *ELF* genes in control of flowering by light quality in petunia. Two *ELF4*-like genes, *PhELF4-1* and *PhELF4-2* with 76% and 70% similarity to orthologues in pepper but low overall similarity to *ELF* genes in *A. thaliana* L., were characterized in petunia and their expression patterns studied under different light qualities. Both genes showed a rhythmic expression pattern and higher expression under B light from light emitting diodes (LED) and W light from fluorescent lamps than under R LED light from LED. For both genes, the expression peaked towards the end of the day, 12 h after start of a 14 h photoperiod. Compared with *PhELF4-2*, *PhELF4-1* expression showed higher amplitude with significantly higher peak expression. As investigated for *PhELF4-1*, such an expression rhythm was kept for two days after transfer of the plants to continuous lighting using B LED, indicating a circadian rhythm. *PhELF4-1* also responded with a phase shift after transfer to short days of an 8 h photoperiod. These results indicate that *PhELF4*-like genes in petunia are under photoperiodic control involving a circadian clock and play a role in signal transduction from one or more B light photoreceptors.

**Keywords:** blue light; light quality; petunia; photoperiodic rhythm; red light; signal transduction

## 1. Introduction

In general, the transition to flowering in plants is well known to be regulated by the light conditions. In this respect, photoperiod is a crucial factor for floral induction in both long-day plants such as *Arabidopsis thaliana* L. and short-day plants like *Chrysanthemum*. Not only photoperiod but also the light quality affects induction of flowering in a wide range of plant species.

Photomorphogenesis, including floral induction, occurs in response to different wavelengths absorbed by different photoreceptors such as phytochromes and cryptochromes [1]. By this, the light receptors initiate signaling pathways which affect expression of genes and proteins controlling the flowering time [1,2]. PHYTOCHROME B (PHYB) is known to control the expression of the floral induction gene *FLOWERING LOCUS T* (*FT*) [3] and CRYPTOCHROME 2 (CRY2) stabilizes the CONSTANS (CO) protein, which in turn induces the expression of the *FT* gene [4,5]. On the other hand,

in *Chrysanthemum*, a floral inhibitor, denoted as antiflorigen, was shown to regulate the photoperiodic flowering responses mediated by the signal from PHYTOCHROME [6].

The circadian clock system is involved in photoperiodic control of flowering in plants. A circadian clock is an endogenous time keeping system that is also regulated by signals from photoreceptors [7]. The light input signals from photoreceptors like PHY, CRY and ZEITLUPE (ZTL) provide cues for setting the internal clock timing. PHYB interacts with core clock components such as CIRCADIAN CLOCK ASSOCIATED 1 (CCA1), ELONGATED HYPOCOTYL (LHY), GIGANTEA (GI), TIMING OF CHLOROPHYLL A/B BINDING PROTEIN 1 (TOC1) and EARLY FLOWERING 3 (ELF3) proteins [8]. Output signals from the core clock components contribute to the floral induction under long- or short-day conditions, depending on the type of day length response. Thus, floral induction is affected by several environmental factors including light quality and photoperiod and depends on complex interactions between photoreceptor signals, the circadian clock system and downstream genes encoding proteins regulating flowering [7]. The *ELF4* gene has important roles in circadian regulation [9]. In the long-day plant *A. thaliana*, *elf4* mutants show early flowering and loss of the ability for sensing day length. The ELF4 protein interacts physically with the GI protein that binds to the promoter of the *CO* gene [10].

In *A. thaliana*, the R and far-red (FR) light-sensing receptor, phytochromes and the B light receptors cryptochromes, specifically PHYA, PHYB and CRY2, have important roles in floral induction [3,11]. In this species, R light delays flowering, whereas FR and B light induce early flowering [1,12]. In petunia classified into also a long-day plant, the timing of transition to flowering relies on a complex interaction between temperature, the light integral and photoperiod [13]. In addition, Haliapas et al., [14] reported that high irradiance and FR end-of-day (EOD) treatment induce floral bud formation in petunia. Furthermore, our previous reports [15,16] have shown that the flowering of petunia (*Petunia × hybrida*) is delayed in plants grown under R light from LEDs and accelerated in plants grown under blue LEDs. However, it is still unclear which floral induction- and signal transduction genes are involved in the regulation of flowering in petunia under different light qualities. In a previous study, we found that the expression of the *FLORAL BINDING PROTEIN 28* (*FBP28*), a *SUPRESSOR OF OVEREXPRESSION OF CO 1*-(*SOC1*) like gene, is reduced and induced in petunia plants grown under R and B light from LEDs, respectively [5]. In *A. thaliana*, SOC1 has a role in transmitting a signal from the FT protein to induce floral bud formation. In another previous study of ours, *PehFT*, a *FT*-like gene from petunia, was identified [16] but involvement of *PehFT* in the floral induction control by light quality was not revealed. Petunia is a typical long-day plant showing strong responses to environmental changes, including changes in light quality and irradiance, with respect to floral induction as well as morphogenesis [15]. Improved knowledge on the molecular control of floral induction in the petunia constitutes a valuable addition to the understanding of photomorphogenesis in long-day plants in general, which is currently largely based on studies of *A. thaliana*.

In this study, we aimed to identify circadian clock-associated *ELF4*-like genes using 3'EST high throughput sequencing and to show the regulation of their circadian expression patterns by light quality and photoperiod. Furthermore, we discussed a hypothesis on their possible involvement in the regulation of the timing of floral induction by light quality and photoperiod.

## 2. Materials and Methods

### 2.1. Plant Material and Precultivation

Seeds of *Petunia×hybrida* Vilm. "Baccarat Blue" (Sakata Seed Corporation, Yokohama, Kanagawa, Japan) were sown in a cell tray filled with a commercial growth medium (Metromix350, Sun Gro Horticulture, Vancouver, BC, Canada). The seeds were germinated at 25 °C under a 12 h photoperiod at a photosynthetic photon flux (PPF) of 70 μmol m$^{-2}$ s$^{-1}$ at 400–700 nm in a growth cabinet (LH-60FL12-DT, Nippon Medical and chemical instrument Co., Ltd., Osaka, Japan) equipped with fluorescent lamps (FL10-B, Hitachi Appliances, Tokyo, Japan) as light sources. After germination, the seedlings were

transplanted individually to 10 cm pots filled with the same commercial growth medium used for germination. The seedlings were fertilized with a commercial fertilizer once a week (Hyponex, N:P:K = 6:10:5, Hyponex, Osaka, Japan).

*2.2. Experimental Growth Conditions*

When four to five true leaves had emerged at day 21 after sowing, the seedlings were transferred to growth cabinets (20–120 plants per cabinet, depending on experiment) at 25 °C with 8 h, 14 h or 24 h photoperiod (depending on experiment) of different light qualities provided by fluorescent lamps and panels of light emitting diodes (LED); White fluorescent lamp (W) LED (MELOW FL5N, Toshiba Lighting and Technology Co., Kanagawa, Japan), red (R) LED (F4F, Phillips LUMILEDS, San Jose, CA, USA) or blue (B) LED (Q3J, Phillips). The spectra of these light sources were measured with a spectro-radiometer (USB2000, Ocean Optics, Dunedeen, FL, USA) (Supplemental Figure S1). During all experiments, the vertical position of the LED panels was adjusted to provide a PPF of 70 $\mu$mol m$^{-2}$ s$^{-1}$ at the top of the plant canopy. The irradiance was measured with a quantum sensor (LI-190 SA, Li-Cor, Lincoln, NE, USA).

*2.3. Gene Identification by 3'EST High Throughput Sequencing*

Three petunia shoots were sampled from each three plants at day 0, 7 and 14 of the treatments with R or B LED light described above. Each plant shoot was frozen in liquid nitrogen and stocked in the deep freezer (−80 °C). Total RNA extraction from the bulked shoot samples of each light quality treatment were performed using an RNeasy Plant Mini Kit (QIAGEN, Hilden, Germany) and then treated with DNase using an RNase-Free DNase set (QIAGEN, Hilden, Germany). The RNA sample qualities were checked by a Bioanalyzer (Agilent 2100 Bioanalyzer, Agilent Technologies Inc., Santa-Clara, CA, USA). Single strand cDNA was synthesized from 2 $\mu$g total RNA using a T7-Oligo (dt) primer with a T7-promoter. Furthermore, double strand cDNA was synthesized by DNA polymerase and RNase H. Subsequently, RNA amplification from this double strand cDNA was done by T7 RNA polymerase. Thereafter, single strand cDNA was synthesized again from 3 $\mu$g RNA template and random primers, and then double strand cDNA was synthesized using a biotin labeled oligo (dt) primer with a 454B adapter. A 454A adapter was attached into the synthesized double and single strand cDNA and was used as the sequence template DNA after alkalizing treatment. The sequence template DNA was bonded into capture beads and amplified by emulsion PCR. Thereafter, these beads were recovered and counted. An appropriate number of beads was added into a PicoTiterPlate and sequencing was done by a Genome Sequencer FLX system (454 Life Sciences, Branford, CT, USA). Trimming treatments were carried out to remove the adapter sequence and the polyA sequence from all sequence data. After data trimming, the assembly of the sequence data was done by MIRA3 (version 3.2.0) [17].

*2.4. RACE PCR for Isolation of ELF4 Like Genes*

Shoots of petunia plants with five true leaves exposed to a 14 h photoperiod from W fluorescent lamps for 2 days were sampled and total RNA was extracted as described in Section 2.3. As a result of the 3'EST sequencing (2.3), 3 different clones of *EARLY FLOWERING 4* (*ELF4*)-like genes were detected. Two EST fragments of *ELF4*-like genes (*PhELF4-1* and *PhELF4-2*) were extended by 3'and 5' RACE PCR (3'-FULL RACE Core Set, TAKARA BIO INC., Kusatsu, Shiga, Japan). DNA fragments extended by RACE PCR were TA cloned (pGEM-Vector System 1, Promega Co., Madison, Wisconsin, USA) and transformed into competent cells (ECOS Competent *Escherichia coli* DH5$\alpha$, TAKARA BIO INC., Kusatsu, Shiga, Japan) and incubated on LB plates. After inserted colony selection, transformed *E. coli* was incubated in liquid medium and then the plasmid was extracted by a plasmid DNA extraction kit (Plasmid DNA Purification, TAKARA BIO INC., Kusatsu, Shiga, Japan). Subsequently, extracted plasmids were sequenced by a DNA sequencer (PRISM 3130-Avant, Applied Biosystems Co., Foster, CA, USA).

*2.5. Expression Analysis of ELF4-Like Genes and Recordings of Developmental Parameters*

In each of four separate experiments (a–d below), 2–3 leaves from each of 3 petunia plants were collected for RNA extraction per sampling time point per light quality treatment (the W, B and/or R light treatments described above; experiment a–c) or photoperiod (experiment d; time courses in experiment b–d) were collected for RNA extraction (Table 1). The sampled leaves from one plant served as a biological replicate, and there were accordingly 3 biological replicates per sampling time per light quality/photoperiod treatment. The sampling started at the 4–5 leaf stage and was performed according to the following schedule:

(a)　Leaves from 3 plants per light quality treatment were sampled after 12 h of lighting the second day under the different light quality treatments of a 14 h photoperiod.

(b)　Leaves from altogether 39 plants were sampled from each light quality treatment at 0, 4, 8, 12, 16, 20, 24, 28, 32, 36, 40, 44 and 48 h after the start of the treatments of a 14 h photoperiod.

(c)　Leaves from altogether 18 plants were sampled from each light quality treatment at 0, 7, 14, 21, 28 and 35 days after the start of the different light quality treatments, with harvest 12 h after the start of the daily lighting of a 14 h photoperiod.

(d)　Leaves from altogether 57 plants were sampled under each of three photoperiods of 8, 14 and 24 h provided by B LEDs (day/night of 14 h/10 h, 8 h/16 h and 24 h/ 0 h) at 0, 4, 8, 12, 16, 20, 24, 28, 32, 36, 40, 44, 48, 52, 56, 60, 64, 68 and 72 h after the start of the B LED treatment. In addition, after 4 weeks under these photoperiods with B light, plant height, number of leaves per plant, number of lateral shoots per plant and percentage of plants that had at least one fully opened flower, were recorded for 6 plants per photoperiod.

The sampled plant leaves were frozen in liquid nitrogen and kept in a deep freezer at −80 °C. Total RNA was extracted using an RNA extraction kit (RNeasy Plant Mini Kit, QIAGEN, Hilden, Germany). Using the extracted total RNA as the template, cDNA was synthesized with a cDNA synthesis kit (PrimeScript II 1st strand cDNA Synthesis Kit, TaKaRa, Kusatsu, Japan). Real-time quantitative PCR expression analysis of *PhELF4*-like genes and the house keeping gene *ACTIN8* were performed using the quantitative PCR kit (Brilliant II SYBR Green QPCR Master Mix, STRAGAGENE, La Jolla, CA, USA), *PhELF4-1* primers (forward 5′-CAAAAGTCGCTTCTCTCTATTCTG-3′; reverse 5′-TGAGTAGTGGCGCTGATTGT-3′), *PhELF4-2* primers (forward 5′-CGGAAGAATGGCGTA TGAGA-3′; reverse 5′-GCTAGGATGTAGAGGAAGCTGAA-3′), and *ACTIN 8* primers (forward 5′-TGT CCCTATCTACGAGGGTTATGC-3′; reverse 5′-GTTAGGTCACGGCCAGCAA-3′). All quantitative PCR analyses were performed on Mx3000p (STRATAGENE, La Jolla, CA, USA). Three biological samples were analyzed per treatment and each of these were analyzed in triplicate. The expression of *PhELF4-1* and *PhELF4-2* was normalized to the *PhACTIN 8* gene and shown as the delta Ct values. We confirmed the stability of the Ct values of *PhACTIN 8* in each experiment. The average Ct values for PhACTIN 8 ± standard deviation in each of the four experiments were (a) 21.4 ± 0.31, (b) 18.6 ± 0.11, (c) 18.6 ± 0.12, and (d) 22.8 ± 0.81. (The overall results in experiment a, b, c and d are shown in Figures 3–5 in the Results part, respectively).

**Table 1.** Overview of the experiments for studies of *ELF4-1* and *EL4-2* transcript levels in petunia.

| Experiment | Light Quality Treatments | Photoperiod (h) | Harvest Time after Start of the Light Quality/Photoperiod Treatments | Number of Replicates Per Treatment Per Time Point ** | Total Number of Plants Harvested |
|---|---|---|---|---|---|
| a | W, R, B * | 14 | After 12 h light the 2nd day | 3 | 9 |
| b | W, R, B | 14 | 0, 4, 8, 12, 16, 20, 24, 28, 32, 36, 40, 44, 48 h | 3 | 39 |
| c | W, R, B | 14 | 0, 7, 14, 21, 28, 35 days (after 12 h light each day) | 3 | 18 |
| d | B | 8, 14, 24 | 0, 4, 8, 12, 16, 20, 24, 28, 32, 36, 40, 44, 48, 52, 60, 64, 68, 72 h | 3 | 57 *** |

* W = White fluorescent light, B = blue LED light, R = red LED light. ** Each sample consisted of 2–3 leaves from one plant. *** Eighteen additional plants per photoperiod were used for studies of growth parameters (6 plants per photoperiod).

*2.6. Statistical Analyses*

After confirming normality using the Ryan-Joiner test and homoscedasticity by Levene's test, analysis of variance (ANOVA) in the general linear model (glm) mode was carried out using the Minitab 19 software (Minitab Inc, PA, USA) ($p \leq 0.05$). To test for differences between means, Tukey's post hoc test ($p \leq 0.05$) was used (Minitab 19). For the gene expression data, log-transformed data was used in the statistical analyses due to lack of normality or/and homoscedasticity of the nontransformed data, whereas for growth parameters nontransformed data was used. One-way ANOVA glm was used to evaluate the effect of light quality on *ELF4-1* and *ELF4-2* transcript levels in experiment a (one sampling time point) and effect of photoperiod (B light) on developmental parameters in experiment d. Two-way ANOVA was used to assess the effect of light quality on the *ELF4-1* and *ELF4-2* transcript levels during time courses, i.e., with light quality (experiment b, c) or photoperiod (experiment d) and time as factors.

## 3. Results

*3.1. Floral Induction and Light Signal Transduction Genes Expressed in Petunia under Different Light Qualities as Revealed by 3′EST High Throughput Sequencing*

3′EST high throughput sequencing showed that there were EST clones of 167,634 reads in the R LED treatment and 147,129 reads in the B LED treatment. Totally, 288,779 assembled contig fragments and 6737 singlet fragments were detected after filtering and assembly steps. An NCBI-blastx search for these contig and singlet fragments revealed 3143 and 2048 specifically expressed EST clones in the R and B LED treatments, respectively.

Thirty-seven genes related to floral induction and light quality signal transduction were isolated (Table 2). For *APETALA (AP)* and *CO*-like genes, there was no difference in read number between the R and B light conditions. *CRY*-like genes encoding B light receptors did also not show any clear difference between these light quality treatments, but *PHYA*-like genes encoding R-FR light receptors showed higher expression under the B LED treatment as compared to the R LED irradiation. For *ELF4*-like genes, three different clones were identified. One of the EST clones (*PhELF4-1*) had higher number of reads under the B LED exposure than under the R LED (R 1 vs. B 14), whereas another (*PhELF4-2*) showed higher number of reads under the R light than the B light (R 4 vs. B 0). Furthermore, as the results of preliminary qRT-PCR analysis on three clones of *ELF4* like genes using the same bulk samples, we found higher gene expressions under blue light than red light in *PhELF4*-1 and *PhELF4*-2 (Data not shown). From those results, we focused on *PhELF4*-1 and *PhELF4*-2 and tried to get their full length cDNA clones.

**Table 2.** Predicted floral induction related genes in petunia detected by 3′EST high throughput sequencing.

| Molecular Function | Organism | Accession No. | BLAST | | Read No. | |
|---|---|---|---|---|---|---|
| | | | Score | E-Value | Red | Blue |
| APETALA (AP) 2-like protein | *Nicotiana tabacum* | gb\|ACY30435.1\| | 193 | $1.00 \times 10^{-47}$ | 4 | 1 |
| APETALA (AP) 2-like protein | *Nicotiana tabacum* | gb\|ACY30435.1\| | 69.7 | $1.00 \times 10^{-10}$ | 0 | 3 |
| APETALA (AP) 2-like protein | *Ipomoea nil* | gb\|ABN10954.2\| | 85.9 | $6.00 \times 10^{-15}$ | 2 | 3 |
| CONSTANS (CO) interacting protein 2a | *Solanum lycopersicum* | gb\|AAS67369.1\| | 79 | $2.00 \times 10^{-13}$ | 8 | 8 |
| CONSTANS (CO) interacting protein 2b, putative | *Capsicum chinense* | dbj\|BAG16521.1\| | 238 | $5.00 \times 10^{-61}$ | 12 | 13 |
| CONSTANS interacting protein 3 | *Solanum lycopersicum* | gb\|AAS67371.1\| | 192 | $1.00 \times 10^{-47}$ | 50 | 36 |
| CONSTANS interacting protein 5 | *Solanum lycopersicum* | gb\|AAS67373.1\| | 290 | $1.00 \times 10^{-76}$ | 15 | 10 |
| CONSTANS interacting protein 6 | *Solanum lycopersicum* | gb\|AAS67374.1\| | 202 | $2.00 \times 10^{-50}$ | 1 | 3 |
| CONSTANS (CO) -like protein | *Solanum tuberosum* | gb\|ABH09237.1\| | 127 | $4.00 \times 10^{-28}$ | 5 | 0 |
| CONSTANS (CO) -like protein | *Solanum tuberosum* | gb\|ABH09237.1\| | 202 | $2.00 \times 10^{-50}$ | 26 | 15 |
| CONSTANS (CO) -like protein | *Solanum tuberosum* | gb\|ABH09237.1\| | 266 | $10^{-121}$ | 9 | 14 |
| CONSTANS (CO) -like protein | *Allium cepa* | gb\|ACT22759.1\| | 83.6 | $8.00 \times 10^{-15}$ | 0 | 2 |
| Cryptochrome (CRY) 2 | *Nicotiana sylvestris* | gb\|ABB36797.1\| | 197 | $2.00 \times 10^{-48}$ | 3 | 1 |
| Cryptochrome (CRY) DASH | *Solanum lycopersicum* | sp\|Q38JU2.2\| | 162 | $2.00 \times 10^{-38}$ | 2 | 3 |
| Cryptochrome (CRY) 1 | *Nicotiana sylvestris* | gb\|ABB36796.1\| | 90.5 | $1.00 \times 10^{-16}$ | 3 | 6 |
| Cryptochrome (CRY) 1 | *Solanum lycopersicum* | gb\|AAD44161.1\| | 67.8 | $1.00 \times 10^{-9}$ | 0 | 4 |
| Early flowering (ELF) 1 protein, similar | *Nasonia vitripennis* | ref\|XP_001601451.1 | 36.2 | 1.4 | 4 | 2 |
| Early flowering (ELF) 4 protein, putative | *Solanum lycopersicum* | gb\|AAW22881.1\| | 135 | $2.00 \times 10^{-30}$ | 6 | 0 |
| Early flowering (ELF) 4 protein, putative | *Solanum lycopersicum* | gb\|AAW22881.1\| | 173 | $2.00 \times 10^{-41}$ | 16 | 7 |
| Early flowering (ELF) 4 protein, putative | *Solanum lycopersicum* | gb\|AAW22881.1\| | 147 | $7.00 \times 10^{-34}$ | 1 | 14 |
| Early light inducible protein (ELIP) | *Solanum lycopersicum* | gb\|AAS92268.1\| | 85.5 | $3.00 \times 10^{-15}$ | 4 | 0 |
| Early light inducible protein (ELIP) | *Solanum lycopersicum* | gb\|AAS92268.1\| | 99 | $2.00 \times 10^{-19}$ | 1 | 1 |
| Early Tobacco Anther1 | *Nicotiana tabacum* | gb\|AAO43000.1\| | 36.2 | $6.00 \times 10^{-5}$ | 2 | 4 |
| Early Tobacco Anther1 | *Nicotiana tabacum* | gb\|AAO43000.1\| | 133 | $2.00 \times 10^{-29}$ | 5 | 10 |
| Embryonic flower (EMF) 2 | *Solanum lycopersicum* | gb\|ABI99480.1\| | 195 | $2.00 \times 10^{-48}$ | 5 | 5 |
| Floral binding protein (FBP) 13 | *Petunia x hybrida* | gb\|AAK21250.1\| | 327 | $6.00 \times 10^{-88}$ | 7 | 7 |
| Floral binding protein (FBP) 22 | *Petunia x hybrida* | gb\|AAK21253.1\| | 132 | $2.00 \times 10^{-34}$ | 2 | 1 |

**Table 2.** *Cont.*

| Molecular Function | Organism | Accession No. | BLAST | | Read No. | |
|---|---|---|---|---|---|---|
| | | | Score | E-Value | Red | Blue |
| Floral binding protein (FBP) 28 | *Petunia x hybrida* | gb\|AAK21257.1\| | 164 | $3.00 \times 10^{-39}$ | 1 | 2 |
| Flower-specific phytochrome-associated protein phosphate 3 (ATFYPP3) | *Arabidopsis thaliana* | ref\|NP_188632.1\| | 265 | $4.00 \times 10^{-69}$ | 7 | 6 |
| Phytochrome A (PHYA) | *Solanum tuberosum* | sp\|P30733.2\| | 222 | $1.00 \times 10^{-56}$ | 3 | 4 |
| Phytochrome A -associated F-box protein/Empfindlicher im Dunkelroten Licht 1 (EID1), putative | *Ricinus communis* | ref\|XP_002523077.1\| | 274 | $6.00 \times 10^{-72}$ | 9 | 14 |
| ZEITLUPE (ZTL) | *Ipomoea nil* | gb\|ABC25060.2\| | 102 | $4.00 \times 10^{-20}$ | 5 | 4 |
| Phytochrome A (PHYA) | *Solanum tuberosum* | sp\|P30733.2\| | 222 | $1.00 \times 10^{-56}$ | 3 | 4 |
| Phytochrome A -associated F-box protein/Empfindlicher im Dunkelroten Licht 1 (EID1), putative | *Ricinus communis* | ref\|XP_002523077.1\| | 274 | $6.00 \times 10^{-72}$ | 9 | 14 |
| ZEITLUPE (ZTL) | *Ipomoea nil* | gb\|ABC25060.2\| | 102 | $4.00 \times 10^{-20}$ | 5 | 4 |

### 3.2. Identification of Two Full Length ELF4-Like Genes in Petunia

Two full length cDNA sequences of *ELF4* like genes, *PhELF4-1* (604 bp) and *PhELF4-2* (724 bp), were isolated by RACE PCR (Supplemental Figure S2). Both clones have the *DUF1313* protein domain (domain of unknown function), known from *ELF4* genes of other plant species such as *A. thaliana*, pepper (*Capsicum annuum*) and tomato (*Solanum lycopersicum*) (Figure 1). The predicted amino acid sequence of *PhELF4-1* is 76% and 65% similar to the ELF4 proteins of pepper and tomato, respectively. For *PhELF4-2*, the predicted amino acid sequence has 70% and 58% similarity with the corresponding sequences of pepper and tomato, respectively. However, there is no overall significant similarity between *PhELF4-1* and *PhELF4-2* and ELF4 protein in *A. thaliana*.

```
PhELF4-1    1 - - -67 MEGTSNFNRYRQNLAKSQSHKTN------------------------DLSYEEGDSEVWNNFSNNYREVQ 204
PhELF4-2    1 - - -120 MEDTSSLNRHRRTLAKSHSrnaaattnnrrrannSIVEDHDFSMEEGDSEVWNTFSNNYR 299
Pepper           1 MEGTSNFNRHRQTMAKSNSRRTNNHRVRDNSTMETFT---DISMEEGDSEIWNNFSNNFR 57
Tomato           1 MEDT--FKRHRQTLAKPQSLTTDDRR---------------------RPRDLSTEEGNSDVWNNFSNRFR 47

PhELF4-1    205 SVLDRNRLLIQQVNENHQSRTHDSMVQNVGLIQELNGN ISKV ASLYSDFNNDFSTMV 384
PhELF4-2    300 EVQSVLDRNRVLIQQVNENHQSRTHDSMVQNVGLIQELNGN ISKV SSLYSEFNTDFSTMI 479
Pepper       58 QVQSVLDRNRLLIQQVNENHQSRTHDSMVQNVGLIQELNGN ISKV ASIYSDFNTDFTTML 117
Tomato       48 QVQSVLDRNRSLIQQVNENHQSRTTDNMVRNVSLIQELNGN ISKV VSLYSDISTNFSTMF 107

PhELF4-1    385 HQRKN- 390
PhELF4-2    480 HQRKN 494
Pepper      118 HQRKN 122
```

**Figure 1.** The amino acid sequence alignment of the predicted PhELF4-1 and PhELF4-2 proteins in petunia and ELF4 proteins of pepper and tomato. The box indicates the four conserved amino acid residues of the I(S/T/F)(K/R)V-type proteins from the DUF1313 family.

### 3.3. Gene Expression of ELF4-Like Genes in Petunia under Different Light Conditions

Towards the end of the second day of the light quality treatments (12 h into the 14 h photoperiod), the expression level of *PhELF4-1* differed significantly ($p = 0.0001$) in petunia plants exposed to different light quality treatments, with approximately 5 times higher transcript levels under the W and B light than the R LED light (Figure 2). The *ELF4-2* transcript levels did not differ significantly under the different light qualities; there was a slight tendency only of higher expression levels under the W and B light than under the R light.

The diurnal expression levels of *PhELF4-1* and *PhELF4-2* were also measured every 4 h during the first and 2nd day of exposure to the different light quality treatments. For both genes, the transcript levels differed significantly between the light quality treatments ($p = 0.015$ for *ELF4-1*, $p = 0.02$ for *ELF4-2*). In addition, the expression levels of both genes differed between different time points ($p = 0.0001$), and there was a significant interaction between light treatment and time point for *ELF4-1* ($p = 0.048$), but not *ELF4-2* ($p = 0.55$). Under all light sources, both genes generally showed low expression levels in the morning and the middle of the day, followed by increased expression towards the end of the day with peak expression 12 h after the start of the 14 h photoperiod (Figure 3). This was followed by a rapid decline to low expression levels that persisted or decreased further during the remaining night period (Figure 3). For *PhELF4-1*, the amplitude of the peak expression was significantly higher ($p < 0.05$) about three times) under the W fluorescent lamps and the B LED panels than under the R LED panels (Figure 3A). In addition, the expression under the B light became stronger at the 2nd day of the experiment with 6 times higher expression levels under the B as compared to the R light, whereas the expression under the W light was similar the two consecutive days. The *PhELF4-2* transcript level was also higher ($p < 0.05$) under the B and W light than under the R light but the difference (about 3-fold) was similar both days (Figure 3B).

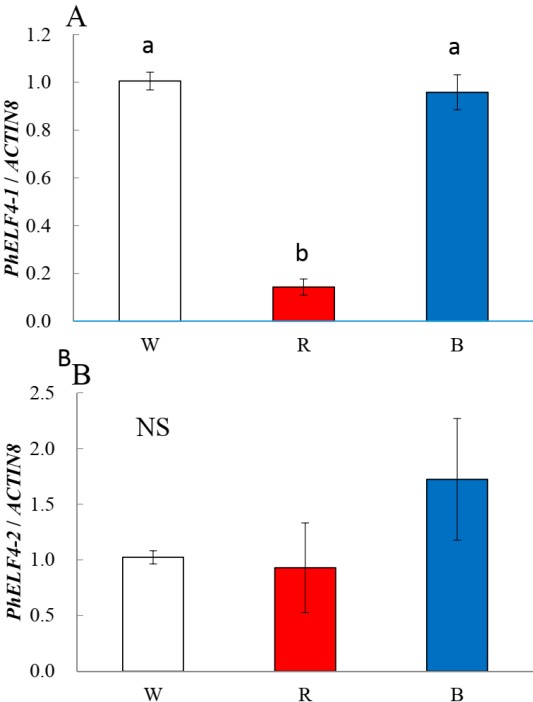

**Figure 2.** Expression of *PhELF4-1* (**A**) and *PhELF4-2* (**B**) in young leaves of petunia, grown under white fluorescent light (W) or blue (B) or red (R) light from light emitting diodes, as analyzed by real-time quantitative PCR. Thamples were harvested 12 h into the 14 h photoperiod on the second day of the treatments and the expression was normalized to the expression of the *PhACTIN 8* gene and shown as mean values relative to the mean value for the W light treatment. The error bars show standard errors ($n = 3$), and different letters within the same column show significant differences among treatments using ANOVA glm followed by Tukey pairwise comparisons ($p < 0.05$). NS = no significant difference between treatments.

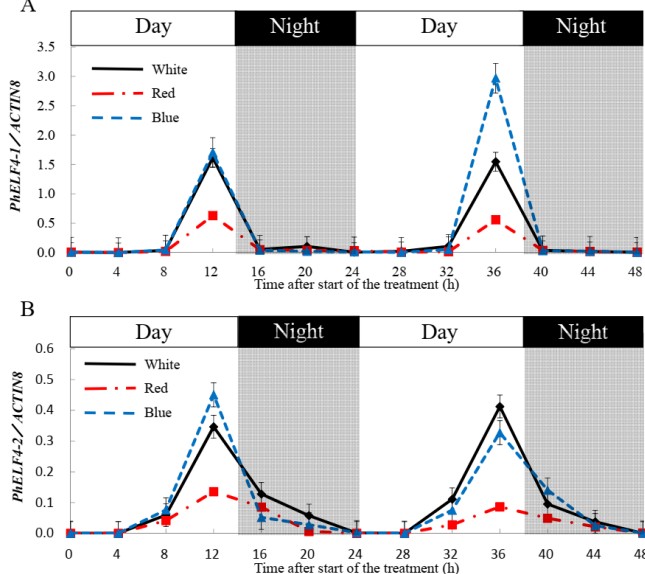

**Figure 3.** Diurnal expression of *PhELF4-1* (**A**) *PhELF4-2* (**B**) in young leaves of petunia plants the first and second day of exposure to white fluorescent light or blue or red light from light emitting diodes under long days of a 14 h photoperiod, as analyzed by real-time quantitative PCR. White and black bars at the top of the graph indicate light (day) and dark (night) periods, respectively. The expression was normalized to the expression of the *PhACTIN 8* gene. The error bars show SE ($n = 3$).

The expression levels of *PhELF4-1* and *PhELF4-2* genes were also measured at different developmental stages of petunia plants grown under the light quality treatments for 35 days (sampling 12 h into the 14 h photoperiod). In this study, the *ELF4-1* expression level differed significantly between the treatments ($p = 0.023$) but this was not the case for *ELF4-2* ($p = 0.888$). For both genes there were significant differences in transcript levels between different time points ($p = 0.0001$). Under all light qualities, the expression levels of *PhELF4-1* and *PhELF4-2* genes increased up to the 7th day of the treatments and then decreased until the end of experiment (Figure 4). At day 7, 14 and 21 the *PhELF4-1* expression levels under the W fluorescent lamps and the B LED panels were significantly higher ($p < 0.05$) as compared to under the R LED panels, i.e., with the largest difference (about twice) in expression level at day 7. Thereafter the expression levels under the W and B light kept being high until the 21th and 28th day of the experiment, respectively, before declining to levels similar to under the R light exposure (Figure 4A). For *PhELF4-2* there was no such significant difference in expression levels among the light qualities and the expression levels were decreased already at the 14th day of the treatments (Figure 4B).

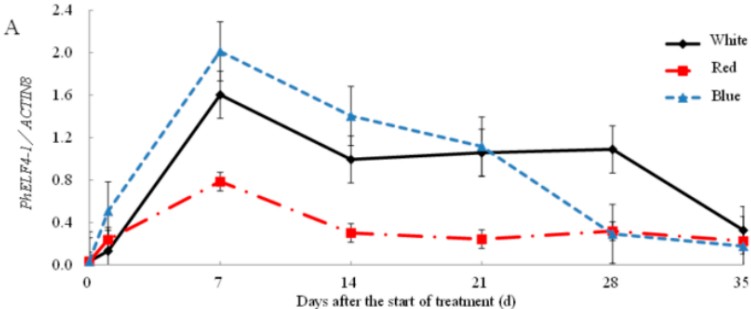

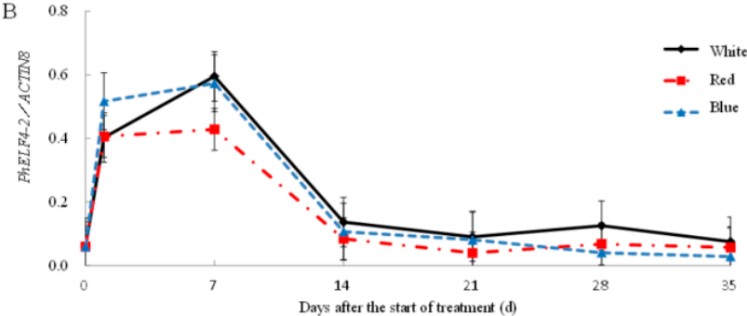

**Figure 4.** Expression of *PhELF4-1* (**A**) *PhELF4-2* (**B**) in young leaves of petunia plants grown under white fluorescent light or blue or red light from light emitting diodes under long days of a 14 h photoperiod, as analyzed by real-time quantitative PCR. All leaves were sampled after 12 h of lighting at each sampling day. The expression was normalized to the expression of the *PhACTIN 8* gene. The error bars show SE ($n = 3$).

The diurnal expression pattern of the *PhELF4-1* gene under the B LED panels was also compared during 3 consecutive days in petunia plants grown under a 14 h photoperiod and plants transferred to a 8 or 24 h photoperiod (i.e., 14 h/10 h, 8 h/16 h or 24 h/0 h day/night, respectively). The transcript level differed significantly between the photoperiods and the different time points, and a significant interaction between these two factors was observed ($p = 0.0001$ in all cases). Under the 14 h photoperiod treatment, the expression peak was observed 12 h into the irradiation period during all 3 days (Figure 5A). However, when the seedlings were moved to a 8 h photoperiod, the timing of the peak expression was shifted. At the 1st day of the experiment, the expression peak was evident after 12 h (2 h into the night), but at the 2nd and 3rd day, the expression peaks were observed after 8 h of irradiation.

Furthermore, when petunia plants were moved into continuous lighting, the expression peak was present at 12 and 36 h after start of the treatment. However, the peak levels gradually decreased, and the peak became less distinct at the 3rd day of the experiment.

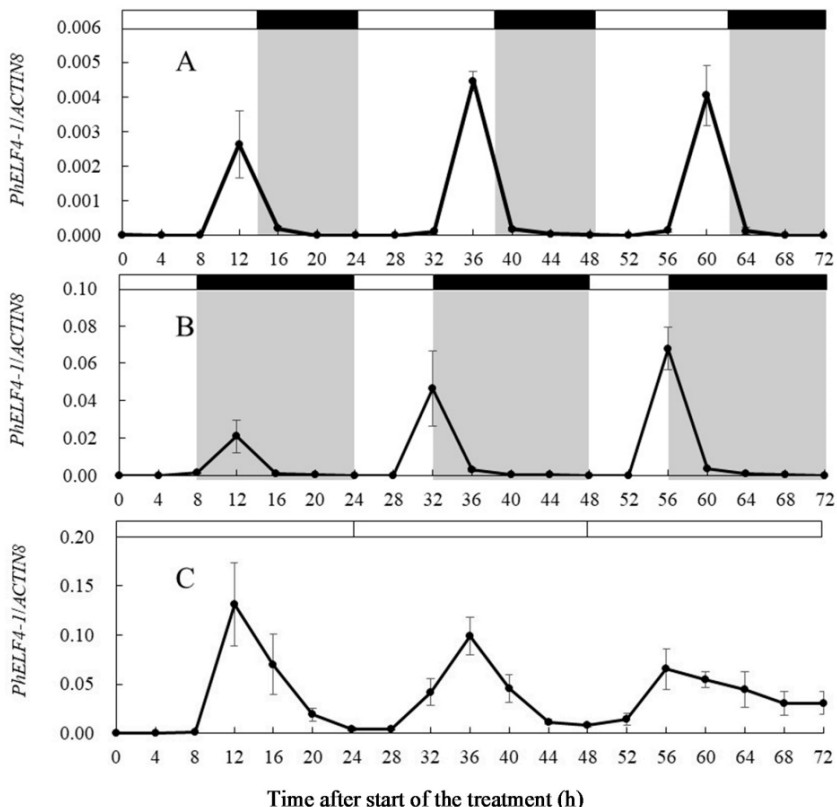

**Figure 5.** Expression of *PhELF4-1* in petunia plants grown under different photoperiods with blue light from light emitting diodes as analyzed by real-time quantitative PCR. (**A**): 14 h light/ 10 h dark, (**B**): 8 h light/ 16 h dark, (**C**): 24 h light/ 0 h dark. White and black bars at the top of each graph indicate light (day) and dark (night) periods, respectively. The expression was normalized to the expression of the *PhACTIN 8* gene. The error bars show SE (*n* = 3).

The plant growth and the percentage of blooming plants were also recorded after 4 weeks under the different photoperiods provided by B LED panels (Table 3). There was a significant effect of photoperiod on all recorded plant growth parameters (plant height and number of lateral shoots $p = 0.0001$, number of leaves $p = 0.004$). The plants were 1.6- and 2-fold taller and the number of leaves was 1.3- and 1.6-fold higher under the continuous lighting compared to the 14 and 8 h photoperiods, respectively. In addition, the number of lateral shoots was higher under the 24 h photoperiod (4 shoots) than under the 14 h photoperiod (0.3 shoots) whereas no lateral shoots were produced under the 8 h photoperiod. Flowers were observed under the long-day conditions of 14 h photoperiod and continuous lighting, i.e., in 75% and 100% of the plants, respectively, but there was no flower development under the 8 h short photoperiod. Furthermore, 24 h photoperiod showed earlier flowering time than 14 h photoperiod, the averaged numbers of days to flowering were 22.8 days under 24 h photoperiod and 28.2 days under 14 h photoperiod, respectively.

**Table 3.** Effects of photoperiod on growth of petunia grown at under blue light emitting diode panels (70 µmol m$^{-2}$ s$^{-1}$ at 25 °C) for 4 weeks.

| Photoperiod (Day/Night) | Plant Height (cm) | No. of Leaves (No. Per Plant) | No. of Lateral Shoots (No. Per Plant) | No. of Days to Flowering | Percentage of Plants with Flowers (%) |
|---|---|---|---|---|---|
| 8 h/16 h | 13.0 b | 19.8 b | 0.0 b | - | 0 |
| 14 h/10 h | 16.4 b * | 25.0 b | 0.3 b | 28.2 b | 75 |
| 24 h/0 h | 26.6 a | 31.5 a | 4.0 a | 22.8 a | 100 |

* Each value shows averaged data (*n* = 6), and different letters within the same column show significant differences among treatments by Tukey's test (*p* < 0.05).

## 4. Discussion

In our previous studies [15,16], flowering of petunia (*Petunia × hybrida*) was shown to be delayed in plants grown under R LEDs and accelerated in plants grown under B LEDs. In addition, the presence of *PehFT* genes was reported, but their function in promotion of flowering is unclear since the expression of *PehFT* was not increased under the B light conditions [16]. In this study, we aimed to identify genes associated with the regulation of flowering time by light quality in petunia.

FR and B light are known to induce flowering in the quantitative long-day plant *A. thaliana* [1,12]. This is associated with absorption of light of different wavelengths by the photoreceptors PHYA, PHYB and CRYs, and their signaling pathways affecting the floral induction genes and proteins controlling flowering time [1,2]. In *A. thaliana* it is known that PHYB controls the expression of the floral induction gene *FT* by sending signals to *PHYTOCHROME AND FLOWERING TIME1* (*PFT1*) [3]. Signals from CRY2 induces flowering by stabilizing the CO protein through CONSTITUTIVE PHOTOMORPHOGENIC1 (COP1) [5]. Furthermore, CRY2 also acts by binding to the CRYPTOCHROME-INTERACTING BASIC-HELIX-LOOP-HELIX1 (CIB1) protein and by directly inducing the expression of *FT* gene. In addition, not only CRY2, but another light-oxygen-voltage (LOV) domain photoreceptor, ZEITLUPE (ZTL), also mediates a signal of B light to prevent the CIB1 protein degradation in *A. thaliana* [18]. In this study, as a result of 3'EST high throughput sequencing, a number of photoreceptor and signal transduction genes were identified including *PHYA*, *CRY*, *CO*, *ZTL*, *SOC1*, *FBP* and *AP*. However, clear effects of different light qualities on expression levels (read numbers) could not be detected for those genes (Table 2). Furthermore, no light quality effect on *PhFT* was detected in this experiment. It may be possible that the bulked sample from different growth stages and young leaves and shoot apices used in this analysis could have resulted in dilution effects on gene expression in a specific organ such as the shoot apex. Furthermore, in this study, *ELF*-like genes were isolated from petunia. It was previously reported that *ELF* genes regulate the circadian rhythm [9,19]. In the circadian clock system in *A. thaliana*, different gene groups show different timing of their expression during the day and nighttime. *CCA1* and *LHY* are expressed in the morning (known as the "morning loop" group I) [20,21], and *LUX ARRHYTHMO (LUX)*, *ELF3* and *ELF4* are activated during the evening (known as the "evening loop") [8,22]. These genes are accordingly related to the circadian clock system in plants [23–25]. Furthermore, some proteins, for instance ELF4, ELF3 and LUX, form a complex body that can receive a signal from photoreceptors such as PHYB for regulation of *GI* expression [7]. Moreover, *FLAVIN-BINDING, KELCH REPEAT, F-BOX1*(*FKF1*) and *GI* are key genes for regulation of *CO* expression [26,27].

In *A. thaliana*, the *ELF4* gene is one of the key genes involved in the core circadian clock for regulating various biological responses such as floral induction [7,28]. In this study, two types of *ELF4*-like genes, *PhELF4-1* and *PhELF4-2* were isolated. Although there is no overall significant similarity between the predicted PhELF4-1 and PhELF4-2 proteins in petunia and the ELF4 protein in *A. thaliana*, the two *ELF4*-like genes include a common domain *DUF1313* that is also included in the *ELF4* gene of *A. thaliana* [28]. Our analysis indicates that *PhELF4* genes belong to the I(S/T/F)(K/R)V-type proteins of the *DUF1313* family (Figure 1). Furthermore, the size of the predicted PhELF4-1 and



PhELF4-2 proteins are 4 to 5 times larger than corresponding genes in other plant species. Thus, it may be possible that *PhELF4-1* and *PhELF4-2* have other functions than other *ELF4*-like genes.

The expression patterns of *PhELF4-1* and *PhELF4-2* increased sharply and showed peak expression towards the end of the day. Such an expression pattern coincided with previously reported expression of *ELF4*-like gene in petunia [29]. The expression patterns of both *PhELF4-1* and *PhELF4-2* resemble the *ELF4* gene expression in *A. thaliana* [30]. The expression of *PhELF4-1* also responded to a change in photoperiod and showed a robust circadian rhythm both under short and long day conditions. The model of the circadian clock in *A. thaliana* is composed of interlocked feedback loops [28]. Li et al. [30] reported that the *ELF4* gene expression is controlled by the coordinated action and interaction between constant positive transcription factors and periodic negative transcription factors. Particularly, CCA1 and LHY are essential for establishing and keeping the circadian change of the *ELF4* gene expression. In addition, the FAR-RED ELONGATED HYPOCOTYL 3 (FHY3) protein associates with the PHYA protein in the signal transduction related to the regulation of the circadian clock. Our results indicate that the *PhELF4-1* and *PhELF4-2* genes probably have *ELF4* function as part of the core system of the circadian clock in petunia. Under short-day conditions using B LED panels as the light source, the petunia plants did not show any flowering during the experimental period of four weeks (Table 3). On the other hand, the signal output from the core circadian clock system including PhELF4 could regulate the floral induction under long days. As described above, photoreceptors such as cryptochrome and phytochrome would then contribute to the induction of the *ELF4* gene by stabilizing the ELONGATED HYPOCOTYL 5 (HY5) protein [7]. The signals from the photoreceptors contribute to sustain robust rhythms of the circadian clock system. Although the different light qualities did not affect the circadian rhythm in expression of *PhELF4-1* and *PhELF4-2* as such, their amplitude of expression was affected. The expression peaks towards the end of day was higher under the W fluorescent lamps and B LED panels than under the R LED panels (Figure 3) and for *PhELF4-1* high expression levels under W and B light were kept during 3–4 weeks (Figure 4A). Thus, it is possible that B light is required to induce increased expression of *ELF4*-like genes in petunia.

Doyle et al. [9] have reported that mutation of *AtELF4* changed the expression patterns of circadian clock genes and caused early flowering under short-day conditions in *A. thaliana*. In addition to the role of *ELF4* genes in *A. thaliana* in photoperiod perception and circadian regulation involved in flowering time control, a role in phytochrome B-mediated de-etiolation has been demonstrated [31], but in other plant species the function of *ELF4* genes is still unknown. Liew et al. [32] reported that the *DIE NEUTRALIS* (*DNE*) locus, an orthologue of *AtELF4*, in garden pea (*Pisum sativum*) inhibited flowering under short-day conditions. However, *PsDNE* may be involved in the circadian clock system and be related to PTOC1, PsLATE BLOOMER 1 (PsLATE1) and other components, but its function may be less central than *AtELF4* [33]. Previously, we found that the flowering of petunia is delayed in plants grown under R LEDs and accelerated in plants grown under B LEDs [15,34]. In this experiment, early flowering under long photoperiods of W and B light is associated with elevated expression of *PhELF4-1* and *PhELF4-2*, suggesting their involvement in acceleration of flowering. Such a role of these genes is antagonistic to the early flowering observed under noninductive short days in the *elf4* mutant of *A. thaliana*. Thus, although there is no direct evidence revealing the relationship between flowering time and *PhELF4-1* and *PhELF4-2* genes, possibly these genes have crucial functions in the circadian clock system-related positive output signal for floral induction in petunia.

## 5. Conclusions

In summary, the gene isolated in this study, *PhELF4-1* and *PhELF4-2*, have robust circadian rhythms and their expression is influenced by photoperiod and light quality with the highest expression under light spectra containing B light. Since these genes include the *DUF1313* domain characteristic of circadian-clock-related *ELF4*-like genes in other plant species, it appears that the *PhELF4-1* and *PhELF4-2* are orthologues of such genes in spite of being considerably longer than and having low overall similarity with *ELF4* in *A. thaliana*.

**Supplementary Materials:** The following are available online at http://www.mdpi.com/2073-4395/10/11/1800/s1, Figure S1: Irradiation and spectral distribution of the light sources used in this study. (A) White fluorescent lamp. (B) Blue light emitting diodes (LED). (C) Red LED, Figure S2: cDNA sequences of PhELF4-1 (A) and PhELF4-2 (B) in petunia.

**Author Contributions:** Conceptualization, N.F.; Investigation, T.S., E.M. and A.T.; Writing—Original draft, N.F. and J.E.O. All authors have read and agreed to the published version of the manuscript.

**Funding:** The MEXT/JSPS KAKENHI Grant Number 22580028 is acknowledged for financial support.

**Acknowledgments:** Thanks are due to Tadashi Hirai, for technical assistance with RACE PCR analysis.

**Conflicts of Interest:** The authors declare no conflict of interest. The funders had no role in the study; in the collection, analyses, or interpretation of data; in the writing of the manuscript, or in the decision to publish the results.

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
