# Peer review of "The Expression of ELF4-Like Genes Is Influenced by Light Quality in Petunia"

_agronomy, doi:10.3390/agronomy10111800_

Round 1

Reviewer 1 Report

This manuscript was well done written.

Overall, experimental design and quality of data were impressive.

Discussion of each data was enough to explain the expression of ELF4-like genes influenced by light quality.

-Reviewer-

Author Response

Thank you very much for your reviewing. I am glad to hear your opinion.

Reviewer 2 Report

This manuscript describes the isolation and expression profile of two petunia genes thought as orthologs of the Elf4 gene from Arabidopsis (PhELF4-1 and PhELF4-2). These two CDNA sequences were identified by 3’ EST sequencing and the genes expression profiling tested the effect of light quality (W= White fluorescent light, B= blue LED light, R = red LED light) on diurnal and circadian profile. In addition, the effect of photoperiod on the expression of the two homologous genes and on the development of the plants was examined. These experiments showed that only one of these genes, PhELF4-1, is showing differential expression under different light quality with higher expression in B vs R light. Moreover, these effects occur under long and not short days, as does the induction to flowering in petunia.

In general, the one thing not clear in this manuscript is, what is the question asked and what are the conclusions? I can understand that EST generated a list of genes related to core circadian clock and floral induction. I can also know that they identified two Elf4-like sequences, and followed their expression, by that showing that it is following a circadian rhythm under CL. Again, that under these later condition plant flower earlier. And, that light quality (B and not R) if changing these both observations. But what is the connection between all these observations, and does it prove that Elf4 regulates flowering?

This lack of reasoning also appears in the description of the results, with no 1-2 sentences that will tell why and how they perfromed these xperiments . For example, the first part of results, including the 3’UTR sequencing.  Besides, the fact that the experiment was not designed initially in a way that will allow to compare expression levels for the transcriptome makes some of the comparison inappropriate. The authors state: “clear effects of different light qualities on expression levels (read numbers) could not be detected for those genes”. How could this be said if there’s no real statistics, and in some genes, there is 14 vs 1 reads in blue vs red light?

One point that is not discussed in the paper is what with all the other Elf4 that were identified? In table 2 there 4 different sequences that were identified in EST as Elf4, and the expression analysis is conducted for two of them. Interestingly, at least one of them seem to show higher read number under R vs B, seemingly opposite to the Elf4-1 they present and discuss. What could be the explanation for that?  

The terminology of the genes related to the circadian clock is not accurate and sometimes including inner contradictions. For example, "Elf genes" are mentioned as genes that regulate the circadian rhythms at one paragraph (p17 row 185), and in the one following it is described as part of the oscillator (p17 row 194). Besides, ZTL is described as a B light receptor (p.17 row 176) but it is not.

Some of the data presented is not all suitable for the main figures to be given in the main text. A cDNA sequence within a manuscript, without having a particular purpose, e.g. compare domains, is more suitable to supplementary data. Similarly, the spectral distribution in figure 1 is supplementary data.

The dataset generated in this paper and the results presented makes this a good start for questioning the role of light quality in modulating clock rhythms and flowering in Petunia. The fact that increased expression of Elf4-1 and 2 is increased by B and not R light may link it to flowering control. However, this paper requires major revision and re-organization between main and supplementary data. Besides, some more accuracy on models and the citation of relevant literature.

Author Response

 A: I would appreciate your reading our article and deep understanding. I changed my some explanations that you pointed out below.

Q1: In general, the one thing not clear in this manuscript is, what is the question asked and what are the conclusions? I can understand that EST generated a list of genes related to core circadian clock and floral induction. I can also know that they identified two Elf4-like sequences, and followed their expression, by that showing that it is following a circadian rhythm under CL. Again, that under these later condition plant flower earlier. And, that light quality (B and not R) if changing these both observations. But what is the connection between all these observations, and does it prove that Elf4 regulates flowering?

A1: Thank you for your question. As you mentioned above, we couldn’t conclude the involvement of these genes in flowering induction. However, in this article, we want to report the new ELF like genes in petunia and they have antagonistic expression for floral induction under blue light compared with Arabidopsis. Furthermore, we hope to show these genes have circadian rhythm and it can respond to the change of light environment as our original points in this article. However, my explanation could induce some misunderstandings on our story.

So I changed our explanation for aims on introduction as shown below;

Line 115-119

“In this study, we aimed to identify circadian clock-associated ELF4-like genes using 3'EST high throughput sequencing, and to show the regulation of their circadian expression patterns by light quality and photoperiod. Furthermore, we discussed a hypothesis on their possible involvement in the regulation of the timing of floral induction by light quality and photoperiod.”

Q2: This lack of reasoning also appears in the description of the results, with no 1-2 sentences that will tell why and how they perfromed these xperiments . For example, the first part of results, including the 3’UTR sequencing.  Besides, the fact that the experiment was not designed initially in a way that will allow to compare expression levels for the transcriptome makes some of the comparison inappropriate. The authors state: “clear effects of different light qualities on expression levels (read numbers) could not be detected for those genes”. How could this be said if there’s no real statistics, and in some genes, there is 14 vs 1 reads in blue vs red light?

A2: I am sorry for lack of some explanations on it. As you pointed out, we couldn’t decide the target genes from only 3’UTR sequencing results. We conducted a preliminary RT-PCR analysis for checking the light quality responses. I added an explanation on it as shown below;  

Line 276-279

“Furthermore, as the results of preliminary qRT-PCR analysis on three clones of ELF4 like genes using the same bulk samples, we found that higher gene expressions under blue light than red light in PhELF4-1 and PhELF4-2 (Data not shown). From those results, we focused on PhELF4-1 and PhELF4-2 and tried to get their full length cDNA clones.”

Q3: One point that is not discussed in the paper is what with all the other Elf4 that were identified? In table 2 there 4 different sequences that were identified in EST as Elf4, and the expression analysis is conducted for two of them. Interestingly, at least one of them seem to show higher read number under R vs B, seemingly opposite to the Elf4-1 they present and discuss. What could be the explanation for that?  

A3: As I showed above, I conducted preliminary qRT-PCR analysis on ELF4 like three genes for focusing on specific light quality response genes. 

Q4: The terminology of the genes related to the circadian clock is not accurate and sometimes including inner contradictions. For example, "Elf genes" are mentioned as genes that regulate the circadian rhythms at one paragraph (p17 row 185), and in the one following it is described as part of the oscillator (p17 row 194). Besides, ZTL is described as a B light receptor (p.17 row 176) but it is not.

Q5: I am sorry for my misunderstandings on some terminological aspects. I changed some explanation and words on those misunderstandings as shown below;

Lin488 “but another light-oxygen-voltage (LOV) domain photoreceptor, ZEITLUPE (ZTL), also mediates a signal of B light to prevent the CIB1 protein degradation in A. thaliana”

Lin507 oscillator -> clock

Line526 oscillation -> circadian change

Q6: Some of the data presented is not all suitable for the main figures to be given in the main text. A cDNA sequence within a manuscript, without having a particular purpose, e.g. compare domains, is more suitable to supplementary data. Similarly, the spectral distribution in figure 1 is supplementary data.

A6: Thank you for your point. I agree with you. I moved those figures to supplemental figures.

Q7: The dataset generated in this paper and the results presented makes this a good start for questioning the role of light quality in modulating clock rhythms and flowering in Petunia. The fact that increased expression of Elf4-1 and 2 is increased by B and not R light may link it to flowering control. However, this paper requires major revision and re-organization between main and supplementary data. Besides, some more accuracy on models and the citation of relevant literature.

A7: Thank you for your deep understanding on our study. As you mentioned above, this study is start line for detail analysis. Now I am planning and promoting to make transgenic line to change expression levels of ELF4 like genes in petunia. We will report those results to show the role of ELF4 like genes on floral induction in petunia. Furthermore, we hope to understand the molecular networks related to flowering regulation of petunia under different light conditions. For those purposes, we will conduct NGS analysis. Thank you for your interesting and understanding for our study again.

Reviewer 3 Report

The purpose of this paper is to investigate the molecular regulation of the photoperiod pathway in petunia, with special attention to light quality. The experiments were soundly made and the paper is clearly written mostly. However, there are still major concerns about this manuscript:

- It seems that important data are missing – Table 1 is supposed to show the expression of several flowering-related genes under different lighting conditions. However, unless I am mistaken, this table didn’t appear in the manuscript. It is difficult to evaluate the presented data – only two ELF genes - without the results obtained for other genes, which would give a totally different concept to the paper.

- I couldn’t find data about the flowering time of the plants – just the flowering percentage. These results are particularly important

 - It was not clear to me how the coding regions of the genes were determined from the 3'EST high throughput sequencing

Minor comment:

Write W or B light uniformly (W-light or W light, in all the paper)

Author Response

We would appreciate your kind cooperation.

-The purpose of this paper is to investigate the molecular regulation of the photoperiod pathway in petunia, with special attention to light quality. The experiments were soundly made and the paper is clearly written mostly. However, there are still major concerns about this manuscript:

Answer :Thank you for your review comments and suggestion.

- It seems that important data are missing – Table 1 is supposed to show the expression of several flowering-related genes under different lighting conditions. However, unless I am mistaken, this table didn’t appear in the manuscript. It is difficult to evaluate the presented data – only two ELF genes - without the results obtained for other genes, which would give a totally different concept to the paper.

Answer : At first, I should apology my mistakes on table numbers. The data was shown in table 2 as the results of 3’EST sequencing results. It shows the read number in genes related into flowering. As based on those values, I choose ELF gene responded to blue light. In this time, I have just focused on this gene, but now I am going to cloning other genes for their detail analysis.

- I couldn’t find data about the flowering time of the plants – just the flowering percentage. These results are particularly important.

Answer : Thank you for your suggestion. I added the flowering timing data on table 3. Furthermore, I added results on it from line 456 to 459.

 - It was not clear to me how the coding regions of the genes were determined from the 3'EST high throughput sequencing

Answer : We couldn’t get coding regions from the results of 3’EST clone. We isolated and cloned the full length cDNA sequence by RACE PCR from those  fragments sequence data of 3’EST sequencing. (Line179-192)

Minor comment:

Write W or B light uniformly (W-light or W light, in all the paper)

Answer : Thank you for your points. I change those and uniform as “B light”.

Round 2

Reviewer 2 Report

No comments. I accept that the authors revised and amended text according to comments in previous review

Author Response

Thank you for your interesting and understanding on our article. I am so happy to hear your opinions and suggestions. I will be going to progress our new research projects on this article. I am looking forward to showing our new results in near future.

Reviewer 3 Report

The authors have made useful changes to improve the manuscript.

Please see the attached file for specific comments.

Author Response

I would like to appreciate your comments and suggestions.

I changed some cite and points as you pointed out. However, at your 1st comment "style", I couldn't find any difference with other part. I checked font size, font style, and other things on style, but all are same with other part. If you find some difference on it, please let me know it again.

Furthermore, some my explanation make you misunderstanding. I changed some my explanations. Please confirm those.

Line 37: I changed expression on it.

Line 38: I added reference No1 for that explanation.

Line 59: I added "receptor" on it.

Line 62: I hope to show the petunia is belonged to long day plant. I changed my expression on it.

Line 211: Thank you for your pointing out. I changed it to italic font.

Figure 1: About full sequence data, as other reviewer pointed out that those data should be moved into supplemental data. So I moved it to supplemental data.

Table 3: I changed the order of data set.

References: I changed it to "References".

I would deeply appreciate your kind suggestion and point again.

This manuscript is a resubmission of an earlier submission. The following is a list of the peer review reports and author responses from that submission.

Round 1

Reviewer 1 Report

The publication I am reviewing is very valuable in terms of current research in the world on the quality of different light spectrum and its impact on plants. The valuable results presented in this publication relate primarily to the effect of different light spectrum on gene expression and their modification.

Detailed review in the attached text of the publication

Author Response

Thank you very much for suggesting some revise points.

Line 92: added plant number that was moved in growth cabinet.

Line 103: When I submit my original file, the letters A,B,C were missing. I set those letters again on this figure.

Line 106: We explained "Three petunia shoots were sampled from each three plants at day 0, 7 and 14 of the treatments with R or B LED light described above."

I would appreciate your advises again.

Reviewer 2 Report

Understanding the mechanisms of flowering in plant species is very important since
these can be related to the production and resistance to abiotic and biotic stress.
In addition, it has been seen that light is a determining factor in these studies and that it can even be very important when predicting epidemics or emerging diseases. That is why I consider the article to be of scientific interest. The designs of the experiments are well planned but I think that some things are necessary:

  1. Scheme of the experiment where the sample collection times, and their purpose are indicated

2.  Is the expression of actin 8 stable between treatments and between times to consider this gene as house keeping? The authors don`t shown analysis of these question and it is crucial for the results.

  1. There is only statistical analysis in same figure. In all experiment have to a correctly statistical analysis, for example:
    1. The authors have been ANOVA, but the data are normally distributed and they are homocedastic?? Because if these two conditions are not true the author can’t use ANOVA either Tukey`s test.
    2. When the authors have analysed measures repeated in time they have to considerer time autocorrelation of data, for example, using glm o glmz with AR1 or other variance-covariance matrix.
    3. The results belong lines 241, 255-267 and 290-299 required statistical analysis.

Author Response

I would appreciate your understanding on our subject and kind suggestion to revise some points.

Our answers are shown below;

Q1:Scheme of the experiment where the sample collection times, and their purpose are indicated.

A1:From line 140-142, we gave an additional explanation. In this experiment, for a better overview of the different experiments where sampling for qPCR analyses was performed, we have added that separate experiments (a-d) were performed.

Q2: Is the expression of actin 8 stable between treatments and between times to consider this gene as house keeping? The authors don`t shown analysis of these question and it is crucial for the results.

A2: In our experiment, we confirmed the stability of the Ct values of PhACTIN8 when we loaded same volume of RNA template. We gave that explanation.

Q3-1: The authors have been ANOVA, but the data are normally distributed and they are homocedastic?? Because if these two conditions are not true the author can’t use ANOVA either Tukey`s test.

A3-1:After confirming normality using the Shaprio-Wilk test and homoscedasticity by the Mendoza's multi sample sphericity test, analysis of variance (ANOVA) was performed.

Q3-2: When the authors have analysed measures repeated in time they have to considerer time autocorrelation of data, for example, using glm o glmz with AR1 or other variance-covariance matrix.

A3-2: We use GLM with AIC check for TukeyHSD test. We state that point in materials and test.

Q3-3:The results belong lines 241, 255-267 and 290-299 required statistical analysis.

A3-3:In the 2.6 Statistical analyse part in Materials and methods we have added that a two-way repeated measurement ANOVA was performed for the time course qPCR results.

Thank you for your advises again.

Round 2

Reviewer 2 Report

The author are ignored all my comments.

Q1:Scheme of the experiment where the sample collection times, and their purpose are indicated.

A1:From line 140-142, we gave an additional explanation. In this experiment, for a better overview of the different experiments where sampling for qPCR analyses was performed, we have added that separate experiments (a-d) were performed.

  • Yes, but nowhere does the number of plants used in each experiment appear. The n of experiments is very important to validate your results and these are high quality

Q2: Is the expression of actin 8 stable between treatments and between times to consider this gene as house keeping? The authors don`t shown analysis of these question and it is crucial for the results.

A2: In our experiment, we confirmed the stability of the Ct values of PhACTIN8 when we loaded same volume of RNA template. We gave that explanation.

  • Yes, the authors gave a explanation but they didn´t show any data o graph about this. The science is not leap of faith.

Q3-1: The authors have been ANOVA, but the data are normally distributed and they are homocedastic?? Because if these two conditions are not true the author can’t use ANOVA either Tukey`s test.

A3-1:After confirming normality using the Shaprio-Wilk test and homoscedasticity by the Mendoza's multi sample sphericity test, analysis of variance (ANOVA) was performed.

  • Shapiro-Wilks test is n<30, the author didn´t show the n of experiment so I can evaluate this. In addition, the authos didn’t show p value and stastistic.
  • Mendoza's multi sample sphericity test is not correct to global homocedasticity, is for homocedasticity within repeated measures. The authos had to analysed homocedasticity and sphericity to use ANOVA.

Q3-2: When the authors have analysed measures repeated in time they have to considerer time autocorrelation of data, for example, using glm o glmz with AR1 or other variance-covariance matrix.

A3-2: We use GLM with AIC check for TukeyHSD test. We state that point in materials and test.

  • TukeyHSD is correct for sample from population with normality distribution so this analysis is wrong.

è The author don´t know what is AIC. The Akaike information criterion (AIC) is an estimator of out-of-sample prediction error and thereby relative quality of statistical models for a given set of data. Given a collection of models for the data, AIC estimates the quality of each model, relative to each of the other models. Thus, AIC provides a means for model selection. For analysis with repeated mesure in time, if the data are not normality distribuited you have to perform GLMZs with with autoregressive term to correct for serial correlation (AR1 or similar). The author have ignored my recommendation.

Q3-3:The results belong lines 241, 255-267 and 290-299 required statistical analysis.

A3-3:In the 2.6 Statistical analyse part in Materials and methods we have added that a two-way repeated measurement ANOVA was performed for the time course qPCR results.

è There isn´t p-values or statistic in the all test. This fact is serious because it reduces the value and scientific rigor of the publication.